# Resurgence of Pertussis in the Gyeongnam Region of South Korea in 2023 and 2024

**DOI:** 10.3390/vaccines12111261

**Published:** 2024-11-08

**Authors:** Hyeokjin Lee, U Jin Cho, Ah-Ra Kim, Sang-Eun Lee, Myungju Lee, Seung Ju Lee, Yu Mi Wi, Sang Hyuk Ma, Dong Han Lee

**Affiliations:** 1Division of Infectious Disease Control & Response, Gyeongnam Regional Center for Disease Control and Prevention, Korea Disease Control and Prevention Agency, Cheongju-si 28159, Republic of Korea; jinny0909@korea.kr (H.L.); ujincho@korea.kr (U.J.C.); ahrakim@korea.kr (A.-R.K.); ondalgl@korea.kr (S.-E.L.); 2Infectious Disease Control Division, Busan City Hall, Busan 47545, Republic of Korea; silk0708@korea.kr; 3Infectious Disease Control Division, Gyeongsangnam-do Provincial Hall, Changwon 51139, Republic of Korea; leeyl26@korea.kr; 4Division of Infectious Diseases, Samsung Changwon Hospital, Sungkyunkwan University School of Medicine, Changwon 51353, Republic of Korea; 5Department of Pediatrics, Changwon Fatima Hospital, Changwon 51394, Republic of Korea; msh6517@hanmail.net

**Keywords:** pertussis, resurgence, vaccination adherence, adolescent transmission, Korea

## Abstract

**Background/Objectives**: Pertussis continues to pose a substantial global health challenge despite widespread vaccination programs. This study aimed to analyze the epidemiological characteristics of recent pertussis cases in the Gyeongnam region of South Korea. **Methods**: We analyzed 419 pertussis cases in the Gyeongnam region of South Korea between October 2023 and April 2024, using data collected from epidemiological investigation reports, medical records, and interviews with health officials and school staff. **Results**: Our analysis revealed a distinct age distribution pattern with minimal cases in infants (0.2% under 1 year) and young children (7.9% in ages 1–6 years), while school-aged children showed the highest incidence (31.8% in ages 7–9 years, 36.0% in ages 10–12 years). The outbreak pattern demonstrated a progressive shift from preschool and elementary school students to middle and high school students. The time from symptoms onset to diagnosis varied significantly across age groups, with a median of 3.0 days (1.0–6.0) overall; notably, this was longer in high school students at 5.0 days (2.3–8.0) (*p* = 0.023). While 92.4% (365/395) of cases were fully vaccinated, substantial delays were observed in third and fourth dose administration (18.2% and 25.8%, respectively), with considerable timing variability for the fifth and sixth doses. **Conclusions**: Our findings highlight the changing epidemiology of pertussis in South Korea, characterized by an age shift toward older children. These results emphasize the need for enhanced surveillance focusing on school-aged populations and the reassessment of vaccination strategies, particularly regarding booster dose timings and adherence.

## 1. Introduction

Pertussis, also known as whooping cough, is an acute respiratory disease caused primarily by *Bordetella pertussis* [1]. It is classified as a second-class infectious disease that requires mandatory reporting in South Korea. Adolescents or adults infected with pertussis may exhibit mild symptoms, whereas children without sufficient immunity can experience prolonged convulsive coughing, which, in severe cases, can lead to respiratory complications such as atelectasis and bronchopneumonia [1,2]. Since the introduction of the whole-cell pertussis vaccine in the 1940s, the global incidence of pertussis has decreased [3]. However, following its replacement by acellular pertussis vaccines due to safety issues, pertussis epidemics still occur approximately every 3–4 years despite high vaccination coverage [4]. Recently, a substantial increase in the number of pertussis cases has been reported globally. Denmark has also reported an ongoing pertussis epidemic, with infection levels reaching 1586 cases in November 2023, corresponding to an incidence of 321 per 100,000 per year [5]. In the US, a total of 14,569 cases of whooping cough were confirmed by 14 September, with more than five times the 2475 cases reported in 2023 [6]. This phenomenon may be attributed to enhanced surveillance systems, the potential emergence of antibiotic-resistant strains, waning immunity resulting from the current acellular pertussis vaccine, and the emergence of secondary pathogens not targeted by the current pertussis vaccine [7,8,9,10].

A national immunization program in South Korea provides diphtheria/tetanus/pertussis (DTap) vaccine at 2, 4, and 6 months, between 15 and 18 months, and between 4 and 6 years of age. Furthermore, a booster dose of DTap is recommended for children aged 11 to 12 years. Outbreaks of pertussis occurred in 2015 and 2018 in the Gyeongnam region (Busan, Ulsan, and South Gyeongsang provinces) of South Korea, with 77 and 270 cases reported, respectively. During the three years of the COVID-19 pandemic (2020–2022), the average number of pertussis cases remained approximately 10 per year. However, since October 2023, there has been an increase in cases compared with the same period in previous years, with confirmed cases continuing to occur as of February 2024 [11,12]. Therefore, we analyzed the epidemiological characteristics of recent pertussis cases in the Gyeongnam region to develop effective infectious disease response strategies.

## 2. Materials and Methods

### 2.1. Patient Data Collection

We included 419 cases reported between October 2023 and April 2024 to analyze the epidemiological characteristics of pertussis cases in the Gyeongnam region. Data from epidemiological investigation reports, the patient medical records reviewed during field investigations, and interviews with health center staff and school officials were obtained. Through this process, we acquired detailed information on the demographic characteristics of the patients, including age and sex distribution, as well as the time from symptom onset to diagnosis for the different age groups. We gathered data on major clinical symptoms and vaccination status, including the number of doses received and the timing of vaccinations. The time from symptom onset to diagnosis was calculated for each age group. We examined potential transmission routes, especially within educational settings such as schools, kindergartens, and after-school programs. Data on cluster outbreaks, particularly those in educational institutions, were also collected. Furthermore, we collected information on the symptom onset dates of identified transmitter-recipient pairs.

This study was approved by the Ethics Committee of the Korea Disease Control and Prevention Agency (KDCA-2024-08-04). Furthermore, annual and monthly data surrounding the reported pertussis cases in Korea were obtained from the website [12].

### 2.2. Definition

Confirmed cases were defined according to the diagnostic criteria of the Korea Disease Control and Prevention Agency for pertussis: a person presenting clinical symptoms consistent with pertussis and meet at least one of the following laboratory criteria: (1) the isolation and identification of *Bordetella pertussis* from clinical specimens (nasopharyngeal swabs, nasopharyngeal aspirates, or sputum), or (2) the detection of specific *B. pertussis* genes in these specimens using molecular methods. The South Korean National Immunization Program for pertussis follows a six-dose schedule: three primary doses of the DTap vaccine at 2, 4, and 6 months of age; a fourth dose between 15 and 18 months; a fifth dose at 4–6 years; and a sixth dose of the Tdap vaccine at 11–12 years of age. All doses are provided free of charge through the National Immunization Program. Adherence to the recommended vaccination schedule was assessed based on the following age ranges for each dose: 60–89 days (1st), 120–149 days (2nd), 180–209 days (3rd), 15–18 months plus 30 days (4th), 4–6 years plus 364 days (5th), and 11–12 years plus 364 days (6th). Vaccinations administered within these intervals were categorized as adherent, whereas those administered earlier or later were classified as non-adherent (early or delayed). Missed doses were recorded as unvaccinated for specific doses. The time from symptom onset to diagnosis was defined as the period between the date of initial clinical symptom onset (primarily cough) reported by the patient or guardian during the epidemiological investigation and the date of laboratory confirmation through the PCR testing of *B. pertussis*.

### 2.3. Statistical Analysis

Discrete data were presented as frequencies and percentages, and continuous variables were summarized as the median and interquartile range after testing for the normality of data using the Shapiro–Wilk normality test. Categorical variables were compared using Pearson’s χ^2^ or Fisher’s exact tests. As non-categorical variables did not follow a normal distribution, we employed the Kruskal–Wallis test to compare differences across the five age groups. For multiple pairwise comparisons, Dunn’s test was performed with Bonferroni correction to control the familywise error rate. The Bonferroni-adjusted significance level was set at α = 0.05/n, where n represents the number of pairwise comparisons. All *p*-values were two-tailed, and *p*-values <0.05 were considered statistically significant after adjustment. All statistical analyses were performed using IBM SPSS Statistics, version 27.0 (IBM Corp., Armonk, NY, USA).

## 3. Results

### 3.1. Epidemiological Characteristics of Pertussis in the Gyeongnam Region in 2023 and 2024

Of the 419 patients, 235 (56.1%) were male and 184 (43.9%) were female. In total, 340 cases (81.1%) were reported from urban areas, while 79 cases (18.9%) were from rural areas. By age group, there were 133 (31.8%) patients aged 7–9 years and 151 (36.0%) aged 10–12 years. The elementary school age group accounted for the highest proportion of patients (67.8%). Initially, the incidence was higher among preschool and elementary school students. However, from December 2023, the number of cases increased among middle and high school students who previously had no cases (Table 1). This trend became particularly pronounced in April 2024, when there was a substantial surge in cases across all age groups (Figure 1).

### 3.2. Clinical Characteristics of Pertussis Cases by Age Group in the Gyeongnam Region in 2023 and 2024

Among the 396 confirmed pediatric and adolescent cases, the majority (n = 379, 95.7%) exhibited coughing symptoms. The median time from symptom onset to diagnosis was 3.0 (1.0–6.0) days for all cases, with notable variations across age groups. High school students (16–18 years) had the longest time to diagnose at 5.0 (2.3–8.0) days (*p* = 0.023). While coughing was the predominant symptom across all age groups (overall 95.7%), the differences in prevalence between age groups were not statistically significant (*p* = 0.475). Sputum production was observed in 9.1% of cases, most frequently in high school students (25.0%). Dyspnea was relatively rare, occurring in only 1.0% of all cases (Table 2).

### 3.3. Vaccination Status

Vaccination history was confirmed for 395 patients. The analysis revealed that 92.4% (365) of patients had been fully vaccinated. This rate is 3.2% lower than the national vaccination rate of 95.6% in South Korea as of 2022. Adherence to the recommended vaccination schedule decreased as the dose increased, from 93.4% for the first dose to 71.6% for the fourth dose. However, adherence rates for the fifth and sixth doses remained high (93.7% and 86.4%, respectively). Significant delays in vaccination were found for the third and fourth doses (18.2% and 25.8%, respectively). The timing of the fifth dose administration varied, with 62.2% of patients receiving it at the recommended age of four years, 19.0% at five years, and 15.6% at six years. Additionally, the timing of the sixth dose administration exhibited considerable variability (Table 3).

## 4. Discussion

This study investigated the epidemiological characteristics of pertussis in the Gyeongnam region of South Korea from 2023 to 2024 and revealed a resurgence of cases despite high vaccination coverage. We observed a substantial shift in age distribution towards older children and adolescents during the outbreak, with high incidence in the elementary school age group. The median time from symptom onset to diagnosis was 3.0 (1.0–6.0) days for all cases, with notable variations across age groups. High school students (16–18 years) had the longest time to diagnose at 5.0 (2.3–8.0) days (*p* = 0.023). Notably, 92.4% of the cases occurred in fully vaccinated individuals, raising questions about vaccine effectiveness and waning immunity. These findings indicate a changing epidemiological landscape of pertussis in South Korea, necessitating the re-evaluation of current prevention and control strategies.

The epidemiological analysis of patients with pertussis in the Gyeongnam region showed that the outbreak was initially centered on preschool children and elementary school students before shifting to middle and high school students. This result is consistent with previous studies, which indicate that the diagnosis of pertussis can be delayed in older age groups compared to younger age groups [13]. As reported by Evans et al. [13], the presentation of pertussis in adolescents often differs from the classic whooping cough seen in younger children, potentially leading to lower clinical suspicion among healthcare providers. They also found that prior receipt of antibiotics commonly used to treat respiratory infections was associated with diagnostic delays. Additionally, in the context of South Korea’s educational environment, we observed that the high academic demands and busy schedules of high school students may result in postponed medical visits until symptoms become more severe. Our analysis also revealed variations in clinical manifestations across age groups, with a notably higher rate of sputum production among high school students (25.0%) compared to other age groups (overall 9.1%, *p* = 0.008). This observation correlates with the longer time from symptom onset to diagnosis in high school students [5.0 (2.3–8.0) days versus the overall median of 3.0 (1.0–6.0) days, *p* = 0.023], suggesting that delayed diagnosis may allow for progression to more pronounced respiratory symptoms. While sputum production showed age-related variation, the overall prevalence of cough-related symptoms remained consistently high across all age groups (95.7%, *p* = 0.475), indicating that the fundamental clinical presentation of pertussis remains relatively uniform despite age-specific variations in certain symptoms. A longer diagnostic period among high school students increases the risk of cluster outbreaks. The observed shift in age distribution over time highlights the need for enhanced surveillance and awareness in this age group.

The epidemiological findings from our study reveal a notable age distribution pattern, with only 0.2% of cases in infants under 1 year and 7.9% in children aged 1–6 years, while the highest-burden was observed in school-aged children. This observed age shift represents a significant epidemiological change that can be attributed to multiple factors. First, our finding that 92.4% of cases occurred in fully vaccinated individuals suggests that waning immunity following DTaP vaccination may play a crucial role, with previous studies indicating significant waning after the fourth year of vaccination [14]. The relatively low booster vaccination rate (86.4% for 11–12-year-olds), combined with the variable timing of booster doses, is particularly concerning. The timing of the fifth dose showed considerable variation, with only 62.2% receiving it at the recommended age of four years, while 19.0% received it at five years and 15.6% at six years. This vaccination timing variability likely creates windows of decreased protection in older age groups. Second, increased social mixing patterns among older children and adolescents in educational settings appear to play a significant role in transmission dynamics [15]. Third, the successful implementation of the maternal Tdap vaccination in 2015 has enhanced passive immunity in neonates and effectively reshaped the age distribution of pertussis cases in our region. Additionally, enhanced case findings among older age groups with atypical presentations may have contributed to the observed age shift pattern. In China, there has been a dramatic increase in pertussis cases, with numbers rising from 1512 cases in June 2023 to 97,669 cases in May 2024, including 25 deaths during this period [16]. A pertussis upsurge in China was characterized by an age shift towards older children and adolescents, similar to our observations [17]. Chinese studies identified the ptxP3 macrolide-resistant B. pertussis MT28 clone as a major contributor to this resurgence [18,19]. This clone exhibits both macrolide resistance and potential vaccine escape properties, raising concerns regarding the effectiveness of current treatment and prevention strategies. Japan has maintained relatively stable pertussis control [20]. However, detailed comparisons with Japanese data are limited as recent official statistics are not yet available in the published literature. The situation in South Korea, particularly in our study region, appears to be part of a broader regional resurgence pattern. While our numbers (419 cases) are smaller compared to China’s outbreak, the increase is significant when compared to the average of approximately 10 cases per year during the COVID-19 pandemic period (2020–2022) in our region. This regional comparison suggests that pertussis resurgence is not unique to South Korea but rather represents a broader Asian phenomenon. Given the geographical proximity and human interchange between China and South Korea and the widespread use of macrolide antibiotics in South Korea, the possibility of similar strains circulating in our country cannot be dismissed. The presence of resistant strains could compromise treatment efficacy and facilitate disease transmission.

The high proportion of fully vaccinated individuals (92.4%) raises questions regarding vaccine effectiveness and the potential need for adjusted vaccination strategies. While most patients received their initial vaccinations according to the recommended schedule, we observed noncompliance with later doses, particularly the third and fourth doses, as well as variability in the timing for the fifth and sixth doses. These findings highlight potential areas for improvement in the pertussis vaccination program, particularly in maintaining adherence to the recommended schedule for the third and fourth doses and ensuring the timely administration of the fifth and sixth doses. Our results underscore the importance of achieving high vaccination coverage and ensuring the timely administration of all doses, particularly booster doses, in older children and adolescents. Furthermore, these observations emphasize the need for continued research into the duration of vaccine-induced immunity and the development of potential strategies to enhance long-term protection against pertussis.

Our study has limitations that should be considered when interpreting the results. First, this study relied on clinical data without microbiological analyses. This limitation precludes the direct identification of the genetic characteristics or antibiotic resistance patterns of *B. pertussis* strains, hindering our ability to elucidate the precise causes of pertussis resurgence. Furthermore, the diagnostic method predominantly used in South Korean laboratories, which targets the IS481 insertion sequence, lacks specificity and may lead to the misidentification of other *Bordetella* species, particularly *B. holmesii*, as *B. pertussis* [21,22,23]. This lack of specificity may lead to the potential misidentification of the causative agent, particularly in older children where *B. holmesii* infections may be more prevalent [22]. The increased detection of *B. holmesii* and *B. parapertussis* in other countries [22,23] raises questions about the potential role of other *Bordetella* species in the resurgence of pertussis. Given that *B. holmesii* or *B. parapertussis* lack the cross-protection provided by existing vaccines [22,24], the high rate of infection among vaccinated individuals in our study could be partially attributed to infections by this or other *Bordetella* species against which the current vaccine may not provide protection. Second, the determination of symptom onset dates relied on patient or guardian recall, which may be subject to recall bias, particularly for cases with mild initial symptoms or gradual onset. This recall bias could be more pronounced in older age groups who may be less attentive to early symptoms or attribute them to common respiratory conditions. Third, the high vaccination coverage in our population might result in modified or attenuated symptoms, making early recognition more challenging and potentially leading to underdiagnosis. Healthcare providers might also have lower clinical suspicion for pertussis in vaccinated adolescents, attributing their symptoms to more common respiratory infections. Fourth, the regional healthcare utilization patterns in Gyeongnam, which ranks 10th among 17 metropolitan cities and provinces with a utilization rate of 77.8%, suggest that cases from areas with limited healthcare access might be underrepresented in our surveillance data. These limitations suggest that our findings may underestimate the true burden of pertussis, particularly among adolescents and in regions with lower healthcare accessibility. Lastly, our current study did not comprehensively examine the transmission dynamics within educational institutions or establish specific thresholds for defining cluster outbreaks. These are indeed crucial aspects for understanding pertussis’ epidemiology in school settings. Additional studies would complement our current findings and provide valuable insights for pertussis control in educational settings. Our present study serves as an important foundation for more detailed investigations by identifying the significant role of school-aged children in pertussis transmission.

Based on our findings, several important areas of research are urgently needed in South Korea. Firstly, there is an urgent need to develop and implement more specific and sensitive diagnostic methods capable of differentiating between *Bordetella* species. This improvement in diagnostic techniques can enhance the accuracy of pertussis surveillance and provide a more precise understanding of the epidemiological landscape. Second, the isolation and genotypic analysis of B. pertussis strains should be conducted to confirm the presence of specific genotypes, such as ptxP3 or the MT28 clone, and to track their distribution over time. Third, the systematic antimicrobial susceptibility testing of isolated strains is essential to determine the prevalence and patterns of macrolide resistance. Fourth, the analysis of vaccine antigen genes in circulating strains, compared with those in vaccine strains, is necessary to evaluate potential vaccine escape. Last, comprehensive longitudinal studies investigating the waning of vaccine-induced immunity are imperative. Such research would provide valuable insights into the duration of vaccine protection, including optimal timing for booster doses and the potential need for adjusted vaccination schedules. The implementation of these comprehensive research initiatives could enable a more accurate understanding of the causes of pertussis resurgence in South Korea and inform the development of effective prevention and control strategies.

## 5. Conclusions

Our findings provide the current epidemiology and clinical presentation of pertussis in South Korea. The observed age-related differences in symptom presentation and time to diagnosis highlight the need for age-specific clinical approaches. These results emphasize the importance of the need for more comprehensive microbiological and molecular studies, as well as the importance of reassessing current vaccination strategies and enhancing surveillance methods to effectively control pertussis transmission.

## Figures and Tables

**Figure 1 vaccines-12-01261-f001:**
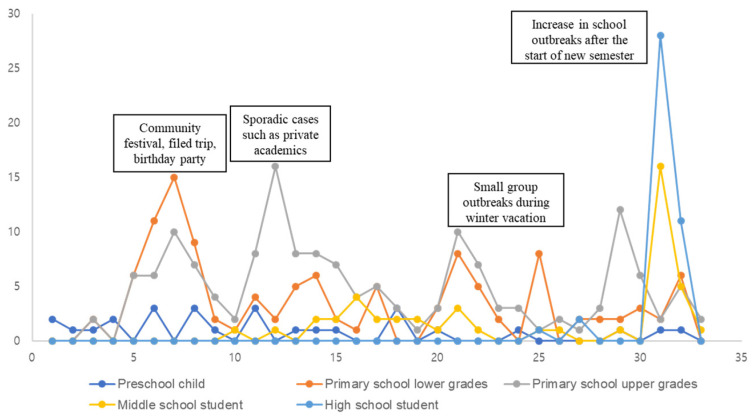
Trends in the occurrence of pertussis cases in the Gyeongnam region from October 2023 to April 2024.

**Table 1 vaccines-12-01261-t001:** General characteristics and monthly occurrence of pertussis cases in the Gyeongnam region (n = 419).

		n (%)	‘23.10.	‘23.11.	‘23.12.	‘24.1.	‘24.2.	‘24.3.	‘24.4.
		419 (100.0)	10	86	70	49	53	29	122
Sex	Male	235 (56.1)	3	49	40	34	29	15	65
	Female	184 (43.9)	10	37	30	15	24	14	57
Age, year	<1	1 (0.2)						1	
	1–6	33 (7.9)	7	8	5	5	2	2	4
	7–9	133 (31.8)	2	57	21	8	21	9	15
	10–12	151 (36.0)	1	18	39	21	23	7	42
	13–15	50 (11.9)			4	10	5	4	27
	16–18	28 (6.7)						1	27
	19 <	23 (5.5)		3	1	5	2	5	7

**Table 2 vaccines-12-01261-t002:** Clinical characteristics of pertussis cases by age group in the Gyeongnam region in 2023 and 2024 (n = 396).

Parameters	Total	* Preschool (0–6 Years)	Lower Elementary (7–9 Years)	Upper Elementary (10–12 Years)	Middle School (13–15 Years)	High School (16–18 Years)	*p*-Value
Total number	396	34 (8.6)	133 (33.6)	151 (38.1)	50 (12.6)	28 (7.1)	
Median time from symptom onset to diagnosis in days (IQR)	3.0 (1.0–6.0)	4.0 (1.0–9.5)	3.0 (1.0–6.0)	3.0 (1.0–6.0)	3.0 (1.0–7.5)	5.0 (2.3–8.0)	0.023
Cough symptoms	379 (95.7)	32 (94.1)	128 (96.2)	147 (97.4)	46 (92.0)	26 (92.9)	0.475
Post-tuissive vomiting	33 (8.3)	6 (17.6)	10 (7.5)	10 (6.6)	2 (4.0)	5 (17.9)	0.058
Sputum	36 (9.1)	6 (17.6)	8 (6.0)	12 (7.9)	3 (6.0)	7 (25.0)	0.008
Dyspnea	4 (1.0)	1 (2.9)	3 (2.2)	0	0	0	0.225

IQR, interquartile range; Data are n (%) unless otherwise stated. * 0: 1, 1–3: 11, 4–6: 22.

**Table 3 vaccines-12-01261-t003:** Vaccination status and adherence to a recommended schedule among pertussis cases in the Gyeongnam region in 2023 and 2024.

Vaccination Dose	Adherence to Schedule, n (%)				Timing of Vaccination (for 5th and 6th Doses), n (%)
	Adherent	Non-Adherent			
		Delayed	Early	Unvaccinated	
1st dose (n = 395)	369 (93.4)	12 (3.1)	14 (3.5)	0	N/A
2nd dose (n = 395)	343 (86.8)	28 (7.1)	13 (3.3)	11 (2.8)	
3rd dose (n = 395)	302 (76.5)	72 (18.2)	11 (2.8)	10 (2.5)	
4th dose (n = 395)	283 (71.6)	102 (25.8)	3 (0.8)	7 (1.8)	
5th dose (n = 395)	370 (93.7)	0	2 (0.5)	23 (5.8)	At 4 years: 239 (62.2%) At 5 years: 73 (19.0%) At 6 years: 60 (15.6%)
6th dose (n = 147)	127 (86.4)	9 (6.1)	2 (1.4)	9 (6.1)	At 11 years: 80 (35.9%) At 12 years: 47 (21.1%) At 13 years: 9 (4.0%)

N/A: not applicable.

## Data Availability

The datasets used in this study are available from the corresponding author upon reasonable request.

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
