# Peer review of "Resurgence of Pertussis in the Gyeongnam Region of South Korea in 2023 and 2024"

_vaccines, 2024, doi:10.3390/vaccines12111261_

Round 1
Reviewer 1 Report
Comments and Suggestions for Authors
The manuscript " Resurgence of Pertussis in the Gyeongnam Region of South Korea in 2023 and 2024" aims to This study aimed to analyze the epidemiological characteristics of recent pertussis cases in Gyeongnam, South Korea, emphasizing transmission patterns and vaccination delays to inform improved control and surveillance strategies. Although the paper presents a certain scientific interest, there are some concerns regarding the data's validity and overall results. Here are some important comments:
1. Please provide an additional interpretation of the trend that is observed in pertussis incidence away from younger children to middle and high school students. Comment on a hypothesis that might explain this pattern including whether or not this may relate to waning immunity, changes in behavior, or increased social mixing.
2. Ensure that the metric of "time from symptom onset to diagnosis" is clarified. Was it to include only the date of clinical symptoms or was laboratory confirmation also a prerequisite for defining the time of diagnosis?
3. Fully describe the methods utilized for ascertaining the routes of transmission in the educational institutions. Detail how potential pairs of transmission were confirmed, and whether genomic sequencing was contemplated to support epidemiological links.
4. Define the thresholds defining a "cluster outbreak" in schools or kindergartens regarding, for instance, the number of cases, period, and spatial limits. This would be a necessary step to ensure replicability and validity when interpreting the outbreak dynamics.
5. If not mentioned, please state whether multiple comparison corrections have been conducted on the statistical analysis, especially in case hypothesis testing has been conducted for more than two age groups, vaccination status, or symptom characteristics. Also, declare threshold adjustments that have been conducted to control the Type I error rate.
6. Account for the statistically significant differences, regarding age groups, in clinical symptoms such as sputum production in high school students.
7. Account for why diagnosis may be longer among the higher age groups, including that of the high school students.
8. It also reports that, compared with the country's vaccination rate, Gyeongnam shows a 3.2% gap. Please elaborate in detail, if possible, on various factors that might be responsible for this difference, including regional healthcare disparities or socioeconomic conditions.
9. Expand Table 3 to include rates of adherence for more granular age categories. Emphasizing such variability may provide clues on which age groups are more susceptible to straying from the recommended schedule; this will help in underlining priority areas of intervention.
10. Discuss the limitations to data collection including recall bias in reporting the onset of symptoms and other age-specific differences in access to health care. Also discuss the potential underreporting among such age groups as high school students, where symptoms may be ignored or misattributed to another cause.
Comments on the Quality of English LanguageThe English language needs more improvement.
Author Response
Dear Sir
November 2, 2024
I really appreciate all reviewers for critical and helpful suggestions, and I feel that the quality of the manuscript has been significantly improved as a result. I provide point-by-point responses to the reviewers' comments. The text in bold signifies the comments made by a reviewer. The authors’ responses appear below each comment.
Modified portions were highlighted in yellow in the manuscript.
We hope that these revisions adequately address the reviewers' concerns. We are grateful for their valuable input, which has helped to enhance the scientific rigor and clarity of our manuscript. We look forward to your feedback on these changes.
Sincerely,
Yu Mi Wi

Reviewer 2 Report
Comments and Suggestions for Authors
Ok so the information confirms past findings, and so the big question...what is next?
Is there a way in which more information could be attained regarding the infected children? Like the SES, are In which types of locations (rural vs. urban vs. suburban) in the Gyeongnam region.?
How about possibly linking these findings to other countries in Asia? What are other countries (Japan) reporting when it coms to pertussis? Is this more of a problem in South Korea?
How about running statistical tests to gauge if cases between the age groups were different in regards to time? Cough symptoms?
Author Response

(The authors gave the same response as above.)

Round 2
Reviewer 1 Report
Comments and Suggestions for Authors
After thoroughly reviewing the revised manuscript and considering the authors' revisions and responses to the referee's comments, I find that the manuscript has been significantly improved. The authors have effectively addressed the concerns, enhancing their study's clarity and scientific rigor. The revisions have clarified the methodology, improved the presentation of results, and strengthened the discussion and conclusions.
Therefore, I believe that the manuscript now meets the standards required for publication in vaccines and recommend that it be accepted for publication.
Thank you for considering my recommendation.
Comments on the Quality of English LanguageThe English language has been enhanced and widely accepted.
Reviewer 2 Report
Comments and Suggestions for Authors
Have covered all of my points.